# Compliance with smoke-free laws in hospitality venues in Ethiopia: A cross-sectional observational study in 10 cities

Wakgari Deressa[1]*, Selamawit Hirpa[1], Terefe Gelibo Argefa[2,3], Yifokire Tefera[1], Selam Abraham Kassa[2], Rachel Kitonyo-Devotsu[2], Winnie Awuor[2], Baharu Zewdie[4], Noreen Dadirai Mdege[2,5,6]

**1** Department of Preventive Medicine, School of Public Health, College of Health Sciences, Addis Ababa University, Addis Ababa, Ethiopia, **2** Development Gateway: an IREX Venture, Washington, DC, United States of America, **3** Mailman School of Public Health, ICAP at Columbia University, Addis Ababa, Ethiopia, **4** Tobacco Control Unit, Ethiopian Food and Drug Authority, Addis Ababa, Ethiopia, **5** Department of Health Sciences, University of York, York, United Kingdom, **6** Centre for Research in Health and Development, York, United Kingdom

* deressaw@gmail.com

**Data Availability Statement:** All relevant data are within the paper and its Supporting Information (S3_dataset).

## Abstract

### Background

The 2019 Ethiopia's comprehensive tobacco control proclamation mandates 100% smoke-free public places and workplaces. Despite the proclamation, compliance remains uncertain, particularly at hospitality venues (HVs). The aim of this study was to evaluate the extent of compliance with smoke-free laws in HVs and to also understand the factors associated with non-compliance.

### Methods

This cross-sectional observational study was conducted in 10 cities in Ethiopia—Addis Ababa, Adama, Assosa, Bahir Dar, Dire Dawa, Gambella, Harar, Hawassa, Jigjiga, and Semera-Logia -between December 5th and 28th, 2022. Data were collected electronically using smartphones, utilizing a pre-tested, standardized checklist and covert observation. The subjects were selected through multi-stage cluster sampling. A total of 1,370 HVs (hotels, restaurants, bars, bars and restaurants, café and restaurants, butcher houses and restaurants, groceries, and nightclubs/ lounges) were observed. Specific and composite compliance indicators were computed for indoor and outdoor spaces. Poisson regression analyses identified factors associated with indoor active smoking and non-compliance. Statistical significance was set at $P<0.05$. Data were analyzed using SPSS version 26.

### Results

Among the 1,370 HVs included in this study, 1,368 had indoor spaces, 327 had both indoor and outdoor spaces, and two had only outdoor spaces. Active smoking was observed in 32.2% (95% CI:30–35) of indoor HVs, with the highest rates in nightclubs/lounges (68.6%) and bars (65.7%). Semera-Logia reported the highest prevalence of active smoking

**Funding:** This study was supported with funding from the Bill and Melinda Gates Foundation (grant number INV-009670). The findings and conclusions contained in the study are those of the authors and do not necessarily reflect the positions and policies of the donor. The funder had no role in study design, data collection, analysis, decision to publish, or preparation of the manuscript.

**Competing interests:** The authors declare no competing interests.

(70.4%). Adherence with 'no smoking' signage was low (35.2%), while ashtrays, lighters, and designated smoking areas (DSAs) were rarely present. Outdoor active smoking was observed in 46.5% (95% CI:41–52) of HVs. Only 12.8% of indoor spaces were fully adherent to the smoke-free law requirements. Venues in Semera-Logia were over twice as likely to have active smoking (adjusted prevalence ratio [aPR]: 2.71; 95% CI: 2.00–3.66) compared to Addis Ababa. Bars and nightclubs/lounges had significantly higher prevalence of active smoking than cafés/restaurants. 'No smoking' signs were associated with reduced prevalence of indoor active smoking (aPR: 0.77; 95% CI: 0.67–0.89), while smoking within a 10-meter range (aPR: 2.67; 95% CI: 2.13–3.32), the presence of lighters (aPR: 1.69; 95% CI: 1.41–2.02), and the sale of tobacco products (aPR: 1.58; 95% CI: 1.34–1.86) were all associated with higher prevalence of indoor active smoking.

## Conclusion

Compliance with 'no active smoking' and adherence to smoke-free laws in HVs remain low, particularly in bars, nightclubs/lounges, and Semera-Logia, with high rates of active smoking both indoors and outdoors. Enhanced enforcement and targeted are needed to educate the public and HV owners about the risks of SHS and the importance of tobacco control laws.

## Introduction

According to the Global Adult Tobacco Survey (GATS) conducted in 2016 in Ethiopia, 3.7% (approximately 2.5 million) of adults (6.2% of men and 1.2% of women) were current cigarette smokers [1], and the smoking prevalence is predicted to rise [2–4]. Exposure of non-smokers to the secondhand tobacco smoke (SHS) is a major concern [5]. From Ethiopia's 2016 GATS, 31% of adults aged 15 years and older who visited restaurants, and 60% who visited bars/nightclubs, were exposed to SHS from tobacco in the 30 days preceding the survey [1]. It has been demonstrated that comprehensive smoke-free legislation effectively reduces SHS exposure among customers, bartenders and workers at hospitality venues (HVs) [6]. Furthermore, it reduces smoking among adolescents and young people [7].

Studies in sub-Saharan Africa (SSA) have indicated that there is a lack of knowledge among HV workers about the smoke-free laws and their requirements, leading to low levels of compliance [8–10]. Furthermore, research in SSA, including Uganda, Ghana, and Kenya, has shown a high prevalence of indoor smoking in HVs [8,11,12], with workers in bars and nightclubs experiencing higher levels of SHS exposure [13,14] and a greater risk of lung cancer mortality [15]. This environment not only endangers workers but also affects vulnerable populations, as children living with a smoker have a significantly higher prevalence of asthma and wheeze, underscoring the urgent need to prevent SHS exposure to safeguard respiratory health [16,17].

Ethiopia enacted a new comprehensive tobacco control law in 2019 (Proclamation No.1112/2019) [18]. The proclamation requires 100% smoke-free indoor public places and workplaces, including HVs. Smoking and the use of tobacco products are prohibited in all indoor workplaces, public places, public institutions, modes of transportation, and communal areas. The proclamation also prohibits designated smoking areas (DSAs) in public places. Additionally, it prohibits the use of any tobacco product in outdoor spaces within a 10-meter radius of any public place or workplace doorway, window, or air intake mechanism, thereby ensuring a comprehensive smoke-free environment. Furthermore, the proclamation mandates

that the owner of the public place or another authorized person must post a 'no smoking' signage and prohibit tobacco use or sale in any part of indoor workplaces. Adherence to the law's requirements involves implementing comprehensive measures designed not only to prevent active smoking in prohibited areas but also to eliminate enabling factors that may undermine a smoke-free environment [18].

A recent scoping review on SHS and smoke-free environments in Ethiopia identified only two studies that assessed compliance with smoke-free laws [19]. One of the two studies was restricted to five government hospitals in Addis Ababa [20] and was conducted prior to the enactment of Proclamation No.1112/2019 [11]. The second study was conducted in 2021 in public places such as those providing food and beverages, schools, health facilities, government offices, youth centers, parks and transit facilities in four regions [10].

Compliance with smoke-free laws is a key pillar of tobacco control policy implementation; however, there is a scarcity of evidence on the extent of compliance with smoke-free laws in Ethiopia, particularly in HVs, which exacerbates the health risks associated with SHS exposure. Therefore, we conducted an assessment of compliance with smoke-free laws in HVs in Ethiopia to provide decision-makers with the evidence they need to strengthen their efforts in protecting non-smokers from exposure to SHS.

In this study, we assessed the implementation of smoke-free laws, focusing on two related concepts: compliance with the smoking ban and adherence to the law's requirements. Compliance refers to the law's effectiveness in preventing active smoking in prohibited areas and serves as a key indicator of its success. While, adherence encompasses how HVs comply with various provisions, such as displaying 'no smoking' signage, removing ashtrays and lighters, banning DSAs, absence of cigarette butts, prohibiting the sale of tobacco products in HVs, and forbidding tobacco use within a 10-meter radius of doorways, windows, or air intake mechanisms of public workplaces. By distinguishing between these concepts, this study aims to clarify the effectiveness of smoke-free legislation.

## Methods

### Study design, setting and population

This cross-sectional study used covert observations to determine compliance with smoke-free laws at HVs in 10 purposively selected major regional and chartered cities in Ethiopia: Addis Ababa, Adama, Assosa, Bahir Dar, Dire Dawa, Gambella, Harar, Hawassa, Jigjiga, and Semera-Logia (where Semera and Logia are two separate nearby cities merged as one study area). Ethiopia had a population of about 120 million in 2022 [21]. The selected cities are characterized by diverse demographic and geographic features, high population density, fast economic development, and high smoking prevalence [1]. They are home to a population of over seven million people. The Tigray Regional State was not included in the study because of security problem at the time of the study.

### Sample size and sampling procedures

We estimated the minimum sample size required for the study using a single proportion sample size formula. We considered a study conducted in Uganda, which found that 82% of the surveyed HVs complied with absence of 'active smoking' in the venues [8]. Based on 95% confidence level, 3% margin of error, design effect of two, and 5% non-response rate, a minimum of 1,300 HVs were required for the study. The sample size was allocated to the study cities and the selected sub-cities, woredas and kebeles within them proportionally to the estimated number of the HVs.

Our sampling procedures followed recommendations for studies on compliance with smoke-free laws [22]. Four of the 10 cities (Addis Ababa, Adama, Bahir Dar and Hawassa) are sub-divided into sub-cities, which are further sub-divided into woredas/kebeles. Two to six sub-cities, with the highest density of registered HVs, were selected from the list of sub-cities in each of these four cities in consultation with the city's tobacco control law enforcement team (S1 Fig). Similarly, from the list of kebeles in each selected sub-city in Adama, Bahir Dar, and Hawassa, two kebeles with the highest density of registered HVs were chosen. In the remaining cities, i.e., Jigjiga, Semera-Logia, Dire Dawa, Harar, Assosa and Gambella, where there are no sub-cities/woredas, we selected 2–6 kebeles/woredas, focusing on those with the highest concentration of HVs.

We obtained initial lists of registered HVs in each selected kebele/woreda from the relevant local offices. Due to incomplete data, we supplemented these lists by conducting on-site mapping, walking street by street within each selected kebele/woreda to identify and list additional HVs. We then used the comprehensive mapped lists to select HVs, either through systematic or random sampling depending on the sub-divisions of each city. For example, systematic sampling was used in cities such as Hawassa and Adama, while random sampling was applied in Addis Ababa and Bahir Dar. In areas where the total number of HVs was small (for example, cities like Harar and Gambella where the identified HVs were particularly less than the allocated sample size), we included all identified HVs. Additionally, for certain HV categories with a limited number of venues, all identified HVs were included in the sample. The assigned sample size for each kebele/woreda was proportionally distributed across the different HV types to ensure representation.

## Eligibility criteria

All selected HVs that were providing services during the time of data collection were included in the study.

## Data collection tools and procedures

This study used a standardized observation checklist to guide covert observations. The observation checklist was developed based on the "How-to-Guide for Conducting Compliance Studies" for Smoke-Free Law [22]. We adapted the checklist considering the provisions of the Ethiopian Tobacco Control Proclamation (1112/2019) [18] and the 2021 tobacco control directive [23]. The tool was initially developed in English, then translated into Amharic, and back-translated into English to ensure consistency. The data collection tool (S1 File) was reviewed and enriched by tobacco control experts in Ethiopia, including those from the Ethiopian Food and Drug Authority (EFDA).

Data were collected by 24 trained data collectors, with one pair of data collectors in each city, except for Addis Ababa, which had three pairs of data collectors. There were 10 field supervisors, one for each city. A three-day training was held in Addis Ababa for all data collectors and supervisors. The data were collected electronically using Open Data Kit (ODK) between December 5th and 28th, 2022, during peak business hours (17:00–24:00).

After scanning the outdoor area and entering the selected HV, the data collectors took a seat as customers, preferably in the main room. In some HVs, they also ordered beverages or food. The specific areas of the HVs that were covertly observed included main entrance, porch, verandah, main room, and two additional rooms, depending on the size of the venue. Other areas observed were lobby areas, corridor areas, hand washing and toilet areas.

Data collectors observed if "no smoking" signage was displayed at the main entrance and inside the venue, whether anyone was smoking a tobacco product indoors or outdoors, and

the presence of ashtrays, lighters, cigarette butts, and shisha equipment. They also noted if any-one was smoking within a 10-meter range of any main door, window, or air intake mechanism while entering or leaving the venue. In addition, they asked the waiter/waitress if the venue sold tobacco products, posing as smokers. If the staff said yes, the data collectors asked to see the available products and their prices. They purchased cigarettes and requested an area to smoke in the venue to check for a DSA, although they later discarded them after leaving the venue. On average, they spent about 30 minutes at each venue. The data collectors entered the data into the smartphone/tablet while they were in or near the HVs without the knowledge of the HV owners or staff members. This approach improved the reliability and validity of the collected data [24].

## Smoke-free compliance and adherence indicators

**Indoor space compliance and adherence.** The following smoke-free specific indicators were used to assess the indoor area compliance and adherence. Compliance was measured by no one being seen smoking a tobacco product in the venue's indoor space (0 = no, 1 = yes). Adherence to the law's requirements was assessed using the following eight indicators: 1) no ashtrays or other instruments used to hold cigarette ash inside (0 = no, 1 = yes), 2) no lighter was visible inside (0 = no, 1 = yes), 3) display of 'no smoking' signage (0 = no, 1 = yes), 4) absence of DSA) (0 = no, 1 = yes), 5) absence of shisha equipment (0 = no, 1 = yes), 6) absence of cigarette butts, 7) no tobacco product sales (0 = no, 1 = yes) and 8) no one was seen smoking tobacco products in the outdoor area within 10-meter range from any door, window, or air intake mechanism (0 = no, 1 = yes). This last indicator was obtained for all HVs, regardless of whether the venue had an outdoor area to serve the public.

**Outdoor space compliance and adherence.** The following smoke-free specific indicators were used to assess the outdoor space compliance and adherence, defined as any area outside of any HV that is not enclosed but serves the public, where smoking is prohibited. Compliance was measured by no one being seen smoking a tobacco product in the venue's outdoor space (0 = no, 1 = yes). Adherence to the law's requirements was assessed using the following four indicators:, 1) no ashtray or other instrument used to hold cigarette ash in the outdoor space (0 = no, 1 = yes), 2) display of 'no smoking' signage in the outdoor space (0 = no, 1 = yes), 3) no cigarette butts seen outdoor, and 4) absence of DSA in the outdoor space (0 = no, 1 = yes).

## Data quality control

Field personnel received training, and the checklists underwent pre-testing to ensure accuracy and reliability. Field supervisors reviewed and checked at least 5% of the data collected by each data collector. Supervisors were also actively involved in data collection alongside data collectors, resolving any inconsistencies through discussions. Each day, collected data were uploaded to a central server at Addis Ababa University, where they underwent daily reviews for completeness and consistency, with immediate feedback from the data manager.

## Operational definitions

Active smoking means someone being in possession or control of a lit tobacco product, including cigarettes, cigars, and shisha, inside or outside of HVs at the time of data collection. An HV is defined as an establishment that is registered under the regulation of the Government of Ethiopia where food and beverages are sold and consumed, namely hotels, restaurants, bars, bars and restaurants, café and restaurant, butcher shop and restaurant, grocery, nightclub and lounges. Further operational definitions used in this study are attached in S2 File.

## Data processing and analysis

Descriptive analysis was used to summarize the characteristics of the HVs, as well as compliance and adherence estimates for each smoke-free indicator. We classified the indoor space of each HV as compliant with the law when no active smoking was detected and as noncompliant when it was detected. In addition, the indoor venues were classified as 'fully adherent', 'highly adherent', 'moderately adherent', and 'poorly adherent' with eight legal smoke-free requirements. An HV was categorized as 'fully adherent' for indoor space if it adhered to eight indicators, 'highly adherent' for adhering with seven indicators, 'moderately adherent' for adhering with 5–6 indicators, and 'poorly adherent' if it met less than five indicators. For outdoor areas, an HV was classified as 'highly adherent ' for outdoor area if it complied with 3–4 indicators, 'moderately adherent' if it met two indicators, and 'poorly adherent' if it met less than two adherence indicators.

Sub-analyses were carried out by study city and HV types. Continuous variables were described using means and standard deviations, while categorical variables were characterized using frequencies and proportions. Chi-squared test was used to compare percentages between the study cities and venue types. The study investigated factors such as city, type of HVs, and the adherence factors associated with indoor active smoking (yes/no) and presence of cigarette butts (yes/no) using Poisson regression analyses with log link function and robust variance [25] and calculated crude prevalence ratios (cPR) and adjusted prevalence ratios (adjPR) with 95% confidence intervals (CIs). Collinearity between cities and HV types was assessed by calculating the Variance Inflation Factor (VIF), and no evidence of multicollinearity was found (VIF < 2). We calculated standard errors, taking into account the complex sample design, which involved adjusting for stratification by city and clustering of HVs at the kebele/cluster level. $P<0.05$ was used for statistical significance. Data were analyzed with SPSS version 26 (IBM SPSS Statistics for Windows, Armonk, NY, USA).

## Ethical considerations

The study protocol for the project titled "Compliance with Smoke-free Laws and Tobacco Advertisement, Promotion and Sponsorship Bans in Ethiopia" was approved by the Institutional Review Board (IRB) of the Ethiopian Public Health Association (EPHA) (Ref. no. EPHA/OG/201/22; dated November 18, 2022). Due to the covert observational nature of the study, informed consent from the HVs or individuals was not obtained. The IRB granted a waiver of informed consent. The observation checklists did not include names, addresses, or any other identifying information, ensuring anonymity, privacy, and data confidentiality. No minors were involved in the study.

## Results

### Characteristics of hospitality venues

Data were collected from a total of 1,370 HVs in 10 cities. About 76% (n = 1,041) of the 1,370 HVs only had indoor spaces, whereas 23.9% (n = 327) had both indoor and outdoor spaces, while two (0.1%) had only outdoor areas. Addis Ababa had the highest number of venues (20.8%), followed by Adama (11.4%), Hawassa (11.3%), and Bahir Dar (10.7%) (Table 1). The remaining six cities were represented by between 7.4% and 7.9% of the HVs. Hotels accounted for 19.3% of all venues, followed by bars and restaurants (17.4%), cafés and restaurants (14.8%), and restaurants (14.7%).

### Indoor active smoking and non-adherence to smoke-free law requirements

Active smoking was observed in indoors of 441 [32.2% (95% CI:30–35)] HVs, with 97.3% (n = 430) of this being from cigarette smoking. Shisha use was rare overall in indoor spaces (1.5%), but it was predominantly observed in indoor areas of nightclubs/lounges (52.4%). Active smoking

**Table 1. Type of hospitality venues by city.**

| City | Venue type, n (%) | | | | | | | | Total, n (%) |
|---|---|---|---|---|---|---|---|---|---|
| | Hotel | Bar and restaurant | Café and restaurant | Restaurant | Grocery | Butcher house and restaurant | Bar | Nightclub /lounge | |
| Addis Ababa | 27 (9.5) | 73 (25.6) | 37 (13.0) | 42 (14.8) | 30 (10.5) | 4 (1.4) | 45 (15.8) | 27 (9.5) | 285 (20.8) |
| Adama | 28 (17.9) | 38 (24.4) | 32 (20.5) | 10 (6.4) | 16 (10.3) | 22 (14.1) | 7 (4.5) | 3 (1.9) | 156 (11.4) |
| Hawassa | 23 (14.9) | 53 (34.4) | 17 (11.0) | 18 (11.7) | 21 (13.6) | 1 (0.6) | 14 (9.1) | 7 (4.5) | 154 (11.3) |
| Bahir Dar | 21 (14.3) | 9 (6.1) | 17 (11.6) | 38 (25.9) | 19 (12.9) | 13 (8.8) | 18 (12.2) | 12 (8.2) | 147 (10.7) |
| Jigjiga | 46 (42.6) | 0 | 22 (20.4) | 22 (20.4) | 7 (6.5) | 11 (10.2) | 0 | 0 | 108 (7.9) |
| Semera-Logia | 23 (21.3) | 1 (0.9) | 47 (43.5) | 37 (34.3) | 0 | 0 | 0 | 0 | 108 (7.9) |
| Harar | 14 (13.3) | 11 (10.5) | 19 (18.1) | 0 | 4 (3.8) | 49 (46.7) | 2 (1.9) | 6 (5.7) | 105 (7.7) |
| Dire Dawa | 18 (17.5) | 21 (20.4) | 2 (1.9) | 5 (4.9) | 29 (28.2) | 8 (7.8) | 12 (11.7) | 8 (7.8) | 103 (7.5) |
| Assosa | 33 (32.0) | 15 (14.6) | 8 (8.8) | 4 (3.9) | 25 (24.3) | 10 (9.7) | 1 (1.0) | 7 (6.8) | 103 (7.5) |
| Gambella | 31 (30.7) | 17 (16.8) | 3 (3.0) | 25 (24.8) | 16 (15.8) | 6 (5.9) | 3 (3.0) | 0 | 101 (7.4) |
| **Total, n (%)** | **264 (19.3)** | **238 (17.4)** | **204 (14.8)** | **201 (14.7)** | **167 (12.2)** | **124 (9.0)** | **102 (7.4)** | **70 (5.1)** | **1,370 (100.0)** |

was the highest in the indoor spaces of HVs in Semera-Logia (70.4%) and Dire Dawa (61.2%). The highest rates of observed active smoking were in the nightclubs/lounges (68.6%) and bars (65.7%). The proportion of HVs that posted 'no smoking signage' in indoor spaces was 35.2% (95% CI:33.0–38.0), with the highest proportion in Semera-Logia (66.7%), whilst this was lowest in Jigjiga (8.3%). Ashtrays, lighters, and shisha equipment were rarely found in the indoor venues across cities, although lighters were observed in 34% of venues in Dire Dawa. The presence of DSAs was generally low, with 8.2% of the HVs in Adama, 5.1% in Semera-Logia, and 2.1% in Bahir Dar.

## Indoor compliance with the law and adherence to the requirements

Table 2 shows compliance and adherence with indoor smoke-free indicators across cities and HVs. There was no active smoking in 67.8% of the HVs. The highest adherence was for the absence of DSAs (98.2%) and shisha equipment (98%). Absence of cigarette butts (64.0%), 'no smoking' signage (64.8%), and no outdoor smoking within 10-meter range of doors or windows (50.2%) were the least adhered to. Adherence with no tobacco product sales was generally high, with Jigjiga showing full adherence, but it was lower in Semera-Logia (46.3%) and among night-clubs/lounges (50%). Most HVs showed high adherence for the absence of ashtrays and lighters, but these enabling aids were more common in bars and nightclubs/lounges. Overall, compliance varied by venue type, with cafés and restaurants showing high adherence, while bars and nightclubs/lounges had the lowest compliance or adherence rates for key indicators.

Only 12.8% of the indoor areas were fully adherent to smoke-free law requirements, and full adherence was the highest in Addis Ababa (22.2%) and cafés and restaurant (23.5%), while poor adherence was the highest in Dire Dawa (25.2%) and nightclubs/lounges (41.4%) (Table 3).

## Outdoor active smoking and non-adherence to smoke-free law requirements

Active smoking was observed in the outdoor space of 46.5% (95% CI:41.0–52.0) of the venues; and this was true for 65.7% of HVs in Semera-Logia and 51.6% of HVs in Adama, compared to 37.9% of HVs in Addis Ababa and 13.8% of HVs in Jigjiga. Active smoking in outdoor

**Table 2. Compliance with the law and adherence to the requirements in indoor areas by city and HV type.**

| City (n = 1,368) | Indoor smoke-free indicator, n (%) | | | | | | | | |
|---|---|---|---|---|---|---|---|---|---|
| | No active smoking | 'No smoking' signage | No ashtray | No lighter | No DSA[a] | No cigarette butts | No shisha equipment | No tobacco product sale | No smoking within 10m |
| Addis Ababa (n = 284) | 209 (73.6) | 152 (53.5) | 271 (95.4) | 255 (89.8) | 284 (100.0) | 211 (74.3) | 276 (97.2) | 279 (98.2) | 131 (46.1) |
| Adama (n = 156) | 117 (75.0) | 50 (32.4) | 145 (92.9) | 148 (94.9) | 143 (91.7) | 130 (83.3) | 154 (98.7) | 139 (89.1) | 100 (64.1) |
| Hawassa (n = 154) | 112 (72.7) | 45 (29.2) | 141 (91.6) | 121 (78.6) | 152 (98.7) | 111 (72.1) | 149 (96.8) | 128 (83.1) | 68 (44.2) |
| Bahir Dar (n = 146) | 113 (77.4) | 19 (13.0) | 137 (93.8) | 130 (89.0) | 143 (97.9) | 111 (76.0) | 141 (96.6) | 88 (60.3) | 111 (76.0) |
| Jigjiga (n = 108) | 100 (92.6) | 9 (8.3) | 108 (100.0) | 108 (100.0) | 108 (100.0) | 90 (83.3) | 108 (100.0) | 108 (100.0) | 93 (86.1) |
| Semera-Logia (n = 108) | 32 (29.6) | 72 (66.7) | 95 (88.0) | 106 (98.1) | 102 (94.4) | 33 (30.6) | 107 (99.1) | 50 (46.3) | 16 (14.8) |
| Dire Dawa (n = 103) | 40 (38.8) | 28 (26.9) | 99 (96.1) | 68 (66.0) | 103 (100.0) | 54 (52.4) | 99 (96.1) | 84 (81.6) | 43 (41.7) |
| Harar (n = 105) | 70 (66.7) | 57 (54.3) | 104 (99.0) | 105 (100.0) | 105 (100.0) | 57 (54.3) | 105 (100.0) | 104 (99.0) | 33 (31.4) |
| Assosa (n = 103) | 69 (67.0) | 33 (32.0) | 103 (100.0) | 90 (87.4) | 102 (99.0) | 30 (22.1) | 102 (99.0) | 86 (83.5) | 31 (30.4) |
| Gambella (n = 101) | 65 (63.7) | 16 (15.8) | 94 (93.1) | 89 (88.1) | 101 (100.0) | 48 (47.5) | 99 (98.0) | 84 (83.2) | 61 (60.4) |
| P-value | <0.001 | <0.001 | <0.001 | <0.001 | <0.001 | <0.001 | 0.260 | <0.001 | <0.001 |
| **Hospitality venue (n = 1,368)** | | | | | | | | | |
| Hotel (n = 264) | 181 (68.6) | 124 (47.0) | 254 (96.2) | 241 (91.3) | 253 (95.8) | 179 (67.8) | 262 (99.2) | 227 (86.0) | 147 (55.7) |
| Bar and restaurant (n = 237) | 173 (73.0) | 96 (40.5) | 229 (96.6) | 216 (91.1) | 233 (98.3) | 157 (66.2) | 234 (98.7) | 219 (92.4) | 112 (47.2) |
| Café and restaurant (n = 204) | 166 (81.4) | 87 (42.6) | 198 (97.1) | 201 (98.5) | 204 (100.0) | 155 (76.0) | 204 (100.0) | 176 (86.3) | 137 (67.2) |
| Restaurant (n = 201) | 148 (73.6) | 52 (25.9) | 190 (94.5) | 195 (97.0) | 198 (98.5) | 128 (63.7) | 201 (100.0) | 168 (83.6) | 113 (56.2) |
| Grocery (n = 167) | 108 (64.7) | 36 (21.6) | 162 (97.0) | 142 (85.0) | 166 (99.4) | 99 (59.3) | 166 (99.4) | 118 (95.9) | 70 (41.9) |
| Butcher house and restaurant (n = 123) | 94 (76.4) | 36 (29.3) | 120 (97.6) | 121 (98.4) | 123 (100.0) | 68 (55.3) | 123 (100.0) | 143 (85.6) | 63 (51.2) |
| Bar (n = 102) | 35 (34.7) | 30 (29.7) | 85 (83.3) | 62 (61.4) | 99 (97.1) | 55 (53.9) | 94 (92.2) | 64 (62.7) | 29 (28.7) |
| Nightclub/lounge (n = 70) | 22 (31.4) | 20 (28.6) | 59 (84.3) | 42 (60.0) | 67 (95.7) | 34 (48.6) | 56 (80.0) | 35 (50.0) | 16 (22.9) |
| P-value | <0.001 | <0.001 | <0.001 | <0.001 | 0.009 | <0.001 | <0.001 | <0.001 | <0.001 |
| **Overall compliance, n (%)** | **927 (67.8)** | **887 (64.8)** | **1297 (94.8)** | **1220 (89.2)** | **1343 (98.2)** | **875 (64.0)** | **1340 (98.0)** | **1150 (84.1)** | **687 (50.2)** |

[a]DSA—Designated Smoking Area

spaces was observed in 71.4% of the bars, 57.1% of the nightclubs/lounges, and 56.9% the restaurants. 'No smoking' signage was observed in the outdoor spaces of 33.1% (95% CI:28.0–38.0) of the HVs, with the highest proportions found in Semera-Logia (59.3%) and Addis Ababa (44.8%). Bars, groceries, nightclubs/lounges, and butcher shops and restaurants had the lowest rates of outdoor 'no smoking' signage. The presence of ashtrays (1.2%) and DSAs (3.6%) was very low in outdoor spaces.

**Table 3. Indoor adherence level with smoke-free requirements by city and HV type.**

| City | Indoor adherence level, n (%)[a] | | | | Total venues |
|---|---|---|---|---|---|
| | **Fully adherent** | **Highly adherent** | **Moderately adherent** | **Poorly adherent** | |
| Addis Ababa | 62 (22.2) | 111 (39.1) | 92 (32.4) | 18 (6.3) | 284 |
| Adama | 28 (17.9) | 66 (42.3) | 51 (32.7) | 11 (7.1) | 156 |
| Hawassa | 28 (18.2) | 44 (28.6) | 51 (33.1) | 31 (20.1) | 154 |
| Bahir Dar | 6 (4.1) | 60 (41.1) | 60 (41.1) | 20 (13.7) | 146 |
| Jigjiga | 8 (7.4) | 76 (70.4) | 24 (22.2) | 0.0 | 108 |
| Semera-Logia | 5 (4.6) | 18 (16.7) | 62 (57.4) | 23 (21.3) | 108 |
| Dire Dawa | 9 (8.7) | 32 (31.1) | 36 (35.0) | 26 (25.2) | 103 |
| Harar | 13 (12.4) | 34 (32.4) | 57 (54.3) | 1 (1.0) | 105 |
| Assosa | 10 (9.7) | 13 (12.6) | 67 (65.0) | 13 (12.6) | 103 |
| Gambella | 5 (5.0) | 25 (24.8) | 58 (57.4) | 13 (12.9) | 101 |
| **Hospitality venue** | | | | | |
| Hotel | 46 (17.4) | 102 (38.6) | 94 (35.6) | 22 (8.3) | 264 |
| Bar and restaurant | 37 (15.6) | 79 (33.3) | 103 (43.5) | 18 (7.6) | 237 |
| Café and restaurant | 48 (23.5) | 92 (45.1) | 53 (26.0) | 11 (5.4) | 204 |
| Restaurant | 18 (9.0) | 80 (39.8) | 86 (42.8) | 17 (8.5) | 201 |
| Grocery | 9 (5.4) | 55 (32.9) | 80 (47.9) | 23 (13.8) | 167 |
| Butcher house and restaurant | 12 (9.8) | 43 (35.0) | 64 (52.0) | 4 (3.3) | 123 |
| Bar | 4 (3.9) | 18 (17.6) | 48 (47.1) | 32 (31.4) | 102 |
| Nightclub/lounge | 1 (1.4) | 10 (14.3) | 30 (42.9) | 29 (41.4) | 70 |
| **Total, n (%)** | **175 (12.8)** | **479 (35.0)** | **558 (40.8)** | **156 (11.4)** | **1,368** |

[a]Fully adherent: Adherence with eight indicators; Highly adherent: Adherence with seven indicators; Moderately adherent: Adherence with 5–6 indicators; Poorly adherence: Adherence with ≤4 indicators.

## Outdoor compliance with the law and adherence with requirements

Table 4 shows that there was no active smoking for 53.5% of the outdoor spaces. There was high outdoor adherence to the absence of ashtrays (98.8%) and DSAs (96.4%). However, lower adherence rates were observed for 'no smoking' signage (33.1%), and absence of cigarette butts (38.6%). City comparisons show significant variations: Bahir Dar reported 100% compliance for no active smoking and 100% adherence to the absence of ashtrays, but only 8.3% had 'no smoking' signage. Jigjiga demonstrated strong compliance (86.2%) for no active smoking, but there was no 'no smoking' signage observed. Semera-Logia had the lowest rates for no active smoking (34.3%) although it had the highest rate for signage (59.3%). Hotels and cafés also showed relatively high compliance and adherence, unlike bars and nightclubs/lounges, which had very low no active smoking and 'no smoking' signage.

About 39% of outdoor spaces in the HVs were highly adherent to smoke-free law requirements, whereas about 42% were found to be in poor adherence with the law requirements (Table 5).

## Factors associated with indoor active smoking and presence of cigarette butts

Table 6 presents results from a bivariate and multivariable Poisson regression model examining predictors of 'active smoking' and presence of 'cigarette butts' in indoor HVs. The prevalence of active smoking and cigarette butts were significantly higher in Semera-Logia and Dire Dawa, both before and after adjustment. After adjustment, venues in Semera-Logia were over

**Table 4. Smoke-free indicators in outdoor spaces by city and HV type.**

| City (n = 329) | Outdoor smoke-free indicator, n (%) | | | | |
|---|---|---|---|---|---|
| | No active smoking | 'No smoking' signage | No cigarette butts | No ashtray | No DSA[a] |
| Addis Ababa (n = 29) | 18 (62.3) | 13 (44.8) | 17 (58.6) | 27 (96.4) | 29 (100.0) |
| Adama (n = 31) | 15 (48.4) | 7 (22.6) | 22 (71.0) | 31 (100.0) | 26 (83.9) |
| Hawassa (n = 37) | 21 (56.8) | 9 (24.3) | 18 (48.6) | 36 (97.3) | 34 (91.9) |
| Bahir Dar (n = 12) | 12 (100.0) | 1 (8.3) | 10 (83.3) | 12 (100.0) | 12 (100.0) |
| Jigjiga (n = 29) | 25 (86.2) | 0 | 10 (34.5) | 29 (100.0) | 29 (100.0) |
| Semera-Logia (n = 108) | 37 (34.3) | 64 (59.3) | 21 (19.4) | 107 (99.1) | 106 (98.1) |
| Dire Dawa (n = 32) | 18 (56.3) | 3 (9.4) | 17 (53.1) | 32 (100.0) | 32 (100.0) |
| Harar (n = 13) | 7 (53.8) | 1 (7.7) | 1 (7.7) | 13 (100.0) | 13 (100.0) |
| Assosa (n = 21) | 12 (57.8) | 9 (42.9) | 4 (19.0) | 21 (100.0) | 21 (100.0) |
| Gambella (n = 17) | 11 (64.7) | 2 (11.8) | 7 (41.2) | 17 (100.0) | 15 (88.2) |
| *P-value* | <0.001 | <0.001 | <0.001 | 0.844 | 0.003 |
| **Hospitality venue (n = 329)** | | | | | |
| Hotel (n = 102) | 58 (56.9) | 38 (37.3) | 38 (37.3) | 100 (98.0) | 95 (93.1) |
| Bar and restaurant (n = 51) | 29 (58.0) | 12 (23.5) | 27 (52.9) | 51 (100.0) | 48 (94.1) |
| Café and restaurant (n = 63) | 34 (54.0) | 33 (52.4) | 19 (30.2) | 63 (100.0) | 62 (98.4) |
| Restaurant (n = 58) | 25 (43.1) | 21 (36.2) | 22 (37.9) | 57 (98.3) | 57 (98.3) |
| Grocery (n = 20) | 12 (60.0) | 1 (5.0) | 8 (40.0) | 20 (100.0) | 20 (100.0) |
| Butcher house and restaurant (n = 21) | 13 (61.9) | 3 (10.0) | 7 (33.3) | 21 (100.0) | 21 (100.0) |
| Bar (n = 7) | 2 (28.6) | 0 | 3 (42.9) | 7 (100.0) | 7 (100.0) |
| Nightclub/lounge (n = 7) | 3 (42.9) | 1 (14.3) | 3 (42.9) | 7 (100.0) | 7 (100.0) |
| *P-value* | 0.514 | <0.001 | <0.001 | 0.942 | 0.411 |
| **Overall compliance, n (%)** | **176 (53.5)** | **109 (33.1)** | **127 (38.6)** | **325 (98.8)** | **317 (96.4)** |

[a]DSA—Designated Smoking Area

two times more likely to have 'active smoking' (aPR: 2.71; 95% CI: 2.00–3.66), while cigarette butts prevalence was about three times higher (aPR: 2.96; 95% CI: 2.23–3.93) than Addis Ababa. Groceries, bars, and nightclubs/lounges had significantly higher prevalence of active smoking and cigarette butts than cafés/restaurants, both before and after adjustment. 'No smoking' signs were associated with reduced prevalence of 'active smoking' (aPR: 0.77; 95% CI: 0.67–0.89). The presence of lighters was strongly associated with higher prevalence of both active smoking (aPR: 1.69; 95% CI: 1.41–2.02 and cigarette butts (aPR: 1.90; 95% CI: 1.58–2.27), before and after adjustment. The presence of tobacco products for sale (aPR: 1.58; 95% CI: 1.34–1.86) and someone smoking within 10-meter range of air intake (aPR: 2.67; 95% CI: 2.13–3.32) strongly associated with higher prevalence of active smoking, before and after adjustment.

## Discussion

About 68% of HVs in our study complied with smoke-free laws in their indoor spaces, defined as 'no indoor active smoking'. The findings of this study highlight significant discrepancies in adherence to smoke-free laws across various indicators. Adherence was notably low for the presence of cigarette butts (64%) and the 'no smoking' signage (47%). In contrast, adherence was high for the absence of DSAs (98.2%) and shisha equipment (98%), along with the absence of ashtrays (94.8%) and lighters (89.2%). Outdoor compliance regarding active smoking was

**Table 5. Outdoor adherence level with smoke-free laws by city and HV type.**

| City | Outdoor adherence level, n (%)[a] | | | Total venues |
|---|---|---|---|---|
| | Highly adherent | Moderately adherent | Poorly adherent | |
| Addis Ababa | 17 (58.6) | 2 (6.9) | 10 (34.5) | 29 |
| Adama | 22 (71.0) | 3 (9.7) | 6 (19.4) | 31 |
| Hawassa | 18 (48.6) | 2 (5.4) | 17 (45.9) | 37 |
| Bahir Dar | 10 (83.3) | 0.0 | 2 (16.7) | 12 |
| Jigjiga | 10 (34.5) | 0.0 | 19 (65.5) | 29 |
| Semera-Logia | 21 (19.4) | 48 (44.4) | 39 (36.1) | 108 |
| Dire Dawa | 17 (53.1) | 1 (3.1) | 14 (43.8) | 32 |
| Harar | 1 (7.7) | 1 (7.7) | 11 (84.6) | 13 |
| Assosa | 4 (19.0) | 6 (28.6) | 11 (52.6) | 21 |
| Gambella | 7 (41.2) | 2 (11.8) | 8 (47.1) | 17 |
| **Hospitality venue** | | | | |
| Hotel | 38 (37.3) | 22 (21.6) | 42 (41.2) | 102 |
| Bar and restaurant | 27 (52.9) | 4 (7.8) | 20 (39.2) | 51 |
| Café and restaurant | 19 (30.2) | 24 (38.1) | 20 (31.7) | 63 |
| Restaurant | 22 (37.9) | 12 (20.7) | 24 (41.4) | 58 |
| Grocery | 8 (40.0) | 1 (5.0) | 11 (55.0) | 20 |
| Butcher house and restaurant | 7 (33.3) | 2 (9.5) | 12 (57.1) | 21 |
| Bar | 3 (42.9) | 0.0 | 4 (57.1) | 7 |
| Nightclub/lounge | 3 (42.9) | 0.0 | 4 (57.1) | 7 |
| **Total, n (%)** | **127 (38.6)** | **65 (19.8)** | **137 (41.6)** | **329** |

[a]Highly adherent: Adherence with 3–4 indicators; Moderately adherence: Adherence with two indicators; Poorly adherent: Adherence with ≤1indicators.

concerning at only 53.5%, with adherence to signage even lower (33.1%). Only 12.8% of venues were fully adherent in indoor areas.

Active smoking was observed in 32.2% of indoor areas and 49.5% of outdoor spaces across various HVs and cities, which is higher than a previous Ethiopian study from 2021 that reported active smoking in only 7% of bars, restaurants, and cafés, and 10% in hotels [10]. This discrepancy may arise from differences in observation timing, variations in venue types, and the higher prevalence of smoking in the cities included in our study compared to the regions assessed in the previous study. A study in Ghana found active smoking in 34.5% of restaurants and 63.6% of bars [11], while in Uganda, it was 18% in HVs [8]. These variations underscore the inconsistent implementation and enforcement of smoke-free laws across HV types, and within and between countries. The higher prevalence of outdoor smoking in our study raises the hypothesis that smokers may be shifting to outdoor areas in response to indoor smoking bans. Furthermore, it is plausible that the HV managers may encourage outdoor smoking if indoor smoking makes customers uncomfortable. A review of compliance facilitators and barriers highlights that smokers with greater knowledge of smoking harms and supportive attitudes toward bans are more likely to adhere to them, while heavier nicotine dependence and negative attitudes can impede adherence [26].

Although the presence of 'no smoking' signage was low in our study (35% in indoors and 33% in outdoor spaces), it was still higher than a previous study in Ethiopia, which reported only 17% of bars and restaurants, and 29% of hotels posted 'no smoking' signage [10]. In Ghana, the presence of such signage was 49.5% [11], with similar findings in other studies [27,28]. Interestingly, our study found exceptionally high adherence rates regarding the

**Table 6. Bivariate and multivariable Poisson regression predictors of presence of 'active smoking' and 'cigarette butts' in indoor places.**

| City | Active smoking | | Cigarette butts | |
|---|---|---|---|---|
| | cPR (95% CI)[a] | aPR[b] (95% CI) | cPR (95% CI) | aPR (95% CI) |
| Addis Ababa | 1[c] | 1 | 1 | 1 |
| Adama | 0.95 (0.68–1.32) | 1.10 (0.81–1.49) | 0.65 (0.43–0.97) | 0.73 (0.50–1.07) |
| Hawassa | 1.03 (0.75–1.43) | 0.75 (0.58–0.97) | 1.09 (0.79–1.50) | 0.92 (0.70–1.22) |
| Bahir Dar | 0.86 (0.60–1.22) | 0.83 (0.59–1.16) | 0.93 (0.66–1.32) | 1.10 (0.76–1.59) |
| Jigjiga | 0.28 (0.14–0.56) | 0.55 (0.28–1.08) | 0.65 (0.41–1.03) | 1.08 (0.68–1.70) |
| Semera-Logia | 2.67 (2.12–3.35) | 2.71 (2.00–3.66) | 2.70 (2.14–3.41) | 2.96 (2.23–3.93) |
| Dire Dawa | 2.32 (1.81–2.97) | 1.47 (1.15–1.88) | 1.85 (1.39–2.46) | 1.39 (1.07–1.80) |
| Harar | 1.26 (0.91–1.76) | 1.53 (1.10–2.15) | 1.78 (1.33–2.37) | 1.76 (1.29–2.40) |
| Assosa | 1.25 (0.89–1.75) | 0.98 (0.72–1.34) | 2.76 (2.18–3.48) | 2.57 (2.02–3.27) |
| Gambella | 1.35 (0.97–1.87) | 1.51 (1.11–2.07) | 2.04 (1.56–2.68) | 2.27 (1.73–2.97) |
| **Hospitality venue** | | | | |
| Café and restaurant | 1 | 1 | 1 | 1 |
| Hotel | 1.69 (1.21–2.37) | 1.96 (1.51–2.54) | 1.34 (0.99–1.81) | 1.20 (0.92–1.56) |
| Bar and restaurant | 1.45 (1.02–2.07) | 1.86 (1.36–2.56) | 1.41 (1.04–1.90) | 1.61 (1.22–2.13) |
| Restaurant | 1.42 (0.98–2.05) | 1.36 (1.04–1.77) | 1.51 (1.11–2.05) | 1.39 (1.08–1.78) |
| Grocery | 1.90 (1.33–2.70) | 1.99 (1.43–2.76) | 1.70 (1.25–2.30) | 1.53 (1.14–2.06) |
| Butcher house and restaurant | 1.27 (0.83–1.94) | 1.55 (1.02–2.36) | 1.86 (1.36–2.55) | 1.86 (1.36–2.53) |
| Bar | 3.53 (2.56–4.85) | 2.82 (2.06–3.87) | 1.92 (1.39–2.65) | 1.65 (1.21–2.25) |
| Nightclub/lounge | 3.68 (2.65–5.11) | 2.44 (1.76–3.39) | 2.14 (1.53–2.99) | 1.38 (0.97–1.97) |
| **Smoke-free law requirements** | | | | |
| 'No smoking' sign (yes/no) | 0.90 (0.77–1.07) | 0.77 (0.67–0.89) | 0.96 (0.83–1.12) | 0.91 (0.79–1.05) |
| Presence of ashtray (yes/no) | 2.50 (2.13–2.93) | 1.11 (0.92–1.34) | 1.97 (1.65–2.35) | 1.20 (0.97–1.49) |
| Presence of lighter (yes/no) | 3.33 (2.97–3.74) | 1.69 (1.41–2.02) | 2.37 (2.09–2.69) | 1.90 (1.58–2.27) |
| Presence of shisha equipment (yes/no) | 2.75 (2.32–3.27) | 1.01 (0.77–1.34) | 2.02 (1.58–2.59) | 1.27 (0.92–1.76) |
| Presence of DSA (yes/no) | 2.29 (1.77–2.96) | 1.06 (0.77–1.47) | 1.80 (1.33–2.44) | 1.44 (0.97–2.13) |
| Tobacco product sale (yes/no) | 3.25 (2.86–3.68) | 1.58 (1.34–1.86) | 1.97 (1.72–2.23) | 1.03 (0.87–1.21) |
| Smoking within 10m (yes/no) | 4.29 (3.47–5.30) | 2.67 (2.13–3.32) | 2.79 (2.35–3.30) | 1.94 (1.63–2.32) |

[a]Crude Prevalence Ratio and 95% Confidence Interval; [b]Adjusted Prevalence Ratio; [c]Reference

absence of DSAs and shisha equipment, suggesting that smoke-free legislation might be effective in specific contexts where enforcement and public health campaigns are prioritized. However, these findings must be viewed alongside lower compliance rates for key indicators such as absence of active smoking, suggesting that while venues may be successful in removing smoking aids, challenges remain in ensuring comprehensive adherence to all smoke-free law requirements [10,29]. Thus, strategies to maintain high levels of adherence to all aspects of smoke-free legislation are essential to protect public health effectively.

Our study revealed regional differences in compliance with smoke-free laws at HVs in Ethiopia, with higher rates in cities like Addis Ababa, Jigjiga, Adama, and Harar, and lower rates in Dire Dawa and Semera-Logia. These differences may be attributed to variations in the local tobacco control environment and the implementation of smoke-free laws. Compliance rates also varied across venue types, with cafés and restaurants demonstrating the highest adherence, while bars and nightclubs/lounges consistently displayed the lowest. This aligns with findings from other studies, indicating that HVs like bars and nightclubs often show low compliance [30]. In Uganda, compliance was particularly low in bars and restaurants, highlighting ongoing challenges in these venues [31]. Thus, targeted interventions in high-risk venues and

cities where smoking is normalized are essential. Implementing stronger penalties, consistent monitoring, and removing enabling factors such as tobacco product sales and lighters could enhance compliance.

The presence of 'no smoking' signage was significantly associated with reduced prevalence of venues with active smoking in violation of the ban. In contrast, the presence of lighters, tobacco product sales, and outdoor smoking within 10-meter range of indoor air intakes were associated with increased smoking rates and cigarette butt prevalence. This aligns with research from Turkey, where indoor cigarette sales and ashtrays were strongly associated with higher indoor smoking rates and cigarette butt availability [32]. In Greece, ashtrays were strongly associated with increased particulate matter ($PM_{2.5}$) levels, indicating their role in encouraging smoking behaviors [33]. However, our research found that ashtrays did not predict compliance with smoke-free laws. Notably, while the Turkish study indicated that 'no smoking' signage did not associate with reduced smoking prevalence, our study showed a significant association between such signage and decreased prevalence of venues with active smoking and cigarette butts. This finding is supported by other studies [10,28], where a significant association was found between the presence of 'no smoking' signage and reduced prevalence of venues with active smoking. While our study indicates a moderate association between the presence of 'no smoking' signage and the reduced prevalence of venues with active smoking, the results highlight the potential benefit of context-specific enforcement strategies and enhanced visual enforcement, such as prominent 'no smoking' signage, along other measures.

Smoking within a 10-meter radius of indoor air intakes poses a significant risk for SHS exposure in HVs, as outdoor tobacco smoke concentrations can easily reach indoors [34]. Our findings show that smoking within this proximity is associated with increased indoor smoking activity in venues and a higher prevalence of cigarette butts, undermining smoke-free policies designed to safeguard indoor air quality. Studies confirm that outdoor tobacco smoke levels remain hazardous even at distances up to nine meters from the source and can infiltrate indoor environments via air intakes [34,35]. Implementing buffer zones around entrances and ventilation intakes could help mitigate SHS infiltration, ensuring better protection for patrons and staff in smoke-free venues.

In our study, full adherence to smoke-free law requirements was observed in only 12.8% of indoor venues, highlighting the significant challenges in enforcing smoke-free regulations, particularly in bars and nightclubs/lounges, and in cities such as Semera-Logia, where adherence was notably low. These findings are consistent with a recent study across four regions in Ethiopia, which reported 12% full compliance in public places [10]. However, the public places assessed in that study, which included government buildings, educational facilities, and transport hubs, differ from the HVs included in our research. Additionally, full compliance in their study was defined as both the absence of active smoking and adherence to smoke-free law requirements. Similar variability in compliance has been noted in other countries. For example, Ghana reported a 70% compliance rate [36], while Bangladesh showed 58% compliance indoors and only 27% outdoors [29]. Meanwhile, studies from Nepal [28] and India [27] reported compliance rates of 26% and 80% in HVs, respectively. These differences emphasize the variability of compliance across settings and suggest that targeted interventions are essential, particularly in environments where social activities and tobacco use are prevalent.

Strict enforcement significantly boosts compliance with smoking bans, as evidenced by a very high compliance rate of 98% in Punjab, India [27]. High compliance levels are often achieved through robust enforcement infrastructure, regular inspections, and penalties [26,37]. Countries that actively involve local authorities, strong political support for smoke-free legislation, and implement prominent 'no smoking' signage see substantial compliance

improvements [26,37]. Involving local jurisdictions in enforcement, particularly through training for inspections, is crucial for maintaining high compliance rates [37]. This localized involvement enables authorities to address specific community dynamics, enhancing enforcement effectiveness. Efficient smoking ban implementation, including clear rewards and punishments, reduces non-compliance [26]. Conversely, inadequate surveillance leads to higher smoking rates, underscoring the need for consistent monitoring. Furthermore, enabling factors like ashtrays in public spaces contribute to increased smoking rates even after smoke-free laws are enacted [26].

The findings of this study have significant policy implications for enhancing tobacco control efforts in Ethiopia, particularly in cities and HVs with lower compliance. The observed variability in compliance across different locations underscores the necessity of tailoring strategies to fit local contexts and dynamics. Strengthening the implementation of Proclamation No. 1112/2019 [18] and Directive No. 771/2021 [23] is crucial for establishing a robust enforcement framework. Given the high non-compliance with the "no smoking within 10-meter" requirement of the law, the presence of lighters, and the sales of tobacco products, focusing on the enforcement of these factors seems advisable when designing implementation efforts, either through enforcement authorities or public education. Additionally, increasing the visibility and enforcement of 'no smoking' signage can serve as an effective policy measure. Despite the moderate association found in this study, international experience shows that clear signage and local authority involvement significantly boost compliance rates [26,37]. Future research should investigate the underlying factors contributing to the low compliance rates observed in HVs, such as bars and nightclubs/lounges, as well as in Semera-Logia city. This exploration will help identify targeted strategies for enhancing enforcement and adherence to smoke-free laws.

## Strengths and limitations

This study is the first large-scale investigation of HVs across 10 major cities in Ethiopia, ensuring diverse context representations. The use of covert observations enhanced data authenticity, providing accurate compliance assessments. The inclusion of multiple indoor and outdoor smoke-free laws indicators offers a detailed understanding of compliance and adherence levels. However, incomplete lists of registered HVs may have led to sampling bias, and the cross-sectional design limits causal conclusions. Despite these limitations, the study provides valuable data and highlights areas needing targeted efforts to improve compliance with smoke-free laws.

## Conclusions

Compliance with 'no active smoking' and adherence to smoke-free law requirements in HVs remain low, particularly in bars, nightclubs/lounges, and Semera-Logia, with high rates of active smoking both indoors and outdoors. Key factors contributing to non-compliance include 'smoking within 10-meter', the presence of lighters, and the sale of tobacco products. Stronger enforcement and targeted interventions are needed to address the low adherence to smoke-free laws. Collaboration among tobacco control stakeholders and the active engagement of local authorities is essential to educate the public and HV owners about the risks of SHS and foster a healthier community. Enhanced monitoring of compliance and adherence, alongside enhanced enforcement of the 'no smoking within 10-meter' requirement, addressing the presence of lighters, regulating the sales of tobacco products, enforcing 'no smoking' signage, and implementing stricter penalties for violations are crucial. These measures, combined with targeted interventions and collaboration among tobacco control stakeholders, can

significantly enhance tobacco control efforts in Ethiopia. Expanding efforts to include informal venues could also provide a more comprehensive understanding of compliance.

## Supporting information

**S1 Fig. Multistage schematic sampling technique.**
(PDF)

**S1 File. Smoke-free study tool.**
(PDF)

**S2 File. Operational definitions.**
(PDF)

**S3 File. Dataset.**
(DTA)

## Acknowledgments

We would like to thank the study team members from Addis Ababa University School of Public Health, Development Gateway: An IREX Venture, Tobacco Control Data Initiative Ethiopia, Ethiopian Food and Drug Authority, and Regional Tobacco Control Regulatory Offices for their valuable support in this study. We are also grateful to all of our data collectors, supervisors, and coordinators who actively participated in this study.

## Author Contributions

**Conceptualization:** Wakgari Deressa, Selamawit Hirpa, Terefe Gelibo Argefa, Selam Abraham Kassa, Rachel Kitonyo-Devotsu, Winnie Awuor, Noreen Dadirai Mdege.

**Data curation:** Wakgari Deressa, Selamawit Hirpa, Terefe Gelibo Argefa.

**Formal analysis:** Wakgari Deressa.

**Funding acquisition:** Terefe Gelibo Argefa, Selam Abraham Kassa, Rachel Kitonyo-Devotsu, Winnie Awuor, Noreen Dadirai Mdege.

**Investigation:** Wakgari Deressa, Selamawit Hirpa, Yifokire Tefera, Noreen Dadirai Mdege.

**Methodology:** Wakgari Deressa, Selamawit Hirpa, Terefe Gelibo Argefa, Yifokire Tefera, Selam Abraham Kassa, Rachel Kitonyo-Devotsu, Winnie Awuor, Baharu Zewdie, Noreen Dadirai Mdege.

**Project administration:** Wakgari Deressa, Selamawit Hirpa.

**Supervision:** Wakgari Deressa, Selamawit Hirpa, Yifokire Tefera, Baharu Zewdie.

**Writing – original draft:** Wakgari Deressa.

**Writing – review & editing:** Wakgari Deressa, Selamawit Hirpa, Terefe Gelibo Argefa, Yifokire Tefera, Selam Abraham Kassa, Rachel Kitonyo-Devotsu, Winnie Awuor, Baharu Zewdie, Noreen Dadirai Mdege.

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
