## [Decision Letter · Decision Letter 0]

24 Sep 2024

PONE-D-24-30728Compliance with smoke-free laws in hospitality venues in Ethiopia: a cross-sectional observational study in 10 citiesPLOS ONE

Dear Dr. Deressa,

Thank you for submitting your manuscript to PLOS ONE. After careful consideration, we feel that it has merit butdoes not fully meet PLOS ONE’s publication criteria as it currently stands. Therefore, we invite you to submit a revised version of the manuscript that addresses the points raised during the review process.

Specifically, reviewer 2 requests additional details and clarification in the definition of compliance and analysis of adherence.

We look forward to receiving your revised manuscript.

Kind regards,

Jennifer Tucker, PhD

Staff Editor

PLOS ONE

3. Thank you for stating the following financial disclosure: [This study was supported with funding from the Bill and Melinda Gates Foundation (grant number INV-009670). The findings and conclusions contained in the study are those of the authors and do not necessarily reflect the positions and policies of the donor.]. Please state what role the funders took in the study. If the funders had no role, please state: "The funders had no role in study design, data collection and analysis, decision to publish, or preparation of the manuscript." If this statement is not correct you must amend it as needed. Please include this amended Role of Funder statement in your cover letter; we will change the online submission form on your behalf.

Additional Editor Comments (if provided):

Reviewers' comments:

Reviewer's Responses to Questions

**Comments to the Author**

1. Is the manuscript technically sound, and do the data support the conclusions?

Reviewer #1: Yes

Reviewer #2: Yes

2. Has the statistical analysis been performed appropriately and rigorously? 

Reviewer #1: Yes

Reviewer #2: Yes

3. Have the authors made all data underlying the findings in their manuscript fully available?

Reviewer #1: Yes

Reviewer #2: No

4. Is the manuscript presented in an intelligible fashion and written in standard English?

Reviewer #1: Yes

Reviewer #2: Yes

5. Review Comments to the Author

Reviewer #1: Investigating the level of implementation to tobacco control law and regulations is a very important task for health policy and public health researchers. This study as the first large scale study looking at hospitality venue smoke-free regulations in Ethiopia, is a valuable contribution to the area. The design, sampling and procedures used in the assessment are up to par with the existing international literature in the area and were reported clearly and concisely in the manuscript. I have made a few comments and requests in the procedures sections of the paper (can be seen in the attached document) to clarify some points but overall, the lit review, design, methods and results sections are well written. However, the discussion section needs a bit more work in terms of concrete policy change and/or implementation process suggestions. To do so, this section also needs to give a bit of background information about the current state of affairs in terms of implementation reinforcement ( by whom, how, how frequently are these reinforcements carried out?), especially if there is need for concern of an increase in lack of compliance. The discrepancy reported between this study findings and a previous similar study may indicate such a concern and that needs to be discussed a bit more in detail (suggestions as to how is made in the text attached). Yet, overall, a good sound study, reported successfully. Thank you.

Reviewer #2: The paper aims to evaluate the level of compliance with smoke-free laws in hospitality venues (HVs) in Ethiopia. Using the largest sample of venues studied to date, the research investigates the extent to which smoking is absent from both indoor and outdoor HVs in this country. Additionally, it assesses the degree to which these venues adhere to enabling legal requirements, such as displaying signage and suppressing designated smoking areas.

This paper makes a valuable contribution to the international understanding of smoke-free law compliance. It provides a comprehensive assessment in a diverse, under-researched context, thus filling a critical gap in the global literature about low-income countries. The study's strengths lie in its large-scale investigation across multiple cities, the use of covert observations for data accuracy, and the inclusion of both indoor and outdoor compliance indicators. Nevertheless, I offer the following comments in the hope of further strengthening the paper.

My main concern revolves around the concept of "compliance with smoke-free laws," an issue in this paper and in some scientific literature. For me, compliance should solely indicate the extent to which the law succeeds in preventing smoking where it's prohibited (the law's objective).

While the law might include other requirements, like signage, meant to aid in smoking prevention, high adherence to these doesn't guarantee a decrease in smoking. Many factors influence the success of a smoking ban. Therefore, it's crucial to strictly reserve "compliance with the law" to describe the law's effectiveness in banning smoking.

In this regard, the paper deviates from a strict definition of compliance in the following ways:

1. Compliance definition: The paper uses a too-broad definition of compliance, encompassing not only the direct observation of smoking (or lack thereof) but also other factors stipulated by the law, such as the presence of 'no smoking' signage, the absence of ashtrays and lighters, and the absence of designated smoking areas (DSAs). While these factors may undoubtedly contribute to a smoke-free environment, they do not directly measure the core objective of the law, which is to prevent smoking in prohibited areas. As indicated before, one thing is compliance with the aim of the law, and another is adherence to the requirements of the law.

2. Compliance in Outdoor Spaces: If compliance is meant to solely indicate the law’s success in preventing smoking where prohibited, then we need to clarify which outdoor areas are covered by the smoking ban. I understand that the outdoor ban applies only within a 10-meter radius of doorways, windows, or air intakes in public places or workplaces. Thus, observing smoking in outdoor areas of HVs may not violate the law. While the analysis of outdoor smoking is valuable, which I recommend keeping, it shouldn't be considered an assessment of compliance with the core objective of the law.

3. Analysis of adherence to enabling requirements of the law: The authors analyze the “factors associated with indoor smoking and below-average compliance.” Their analysis is, however, muddled by the lack of clarity about the compliance concept.

a. First, Table 6, which examines the prevalence ratios of "indoor active smoking" and "below-average compliance," encounters a methodological challenge due to the inherent relationship between these two variables. As the text notes, the "below-average compliance" indicator is partially calculated by considering the presence of active smoking indoors. As a result, the 95%CI of PRs overlap for most cities and venue types. I suggest that the “below-average compliance” variable be transformed into a “below-average adherence to enabling requirements” excluding active smoking. By the way, I would also exclude the variable related to the use of tobacco products outdoors within 10 meters from any door, window, or air intake. It conceptually belongs to compliance with outdoor requirements, and it is not explicitly about smoking but the use of any tobacco product, including non-combustible (although I understand non-combustible use is not frequent in the country).

b. The paper only analyzes three variables (city, venue type, and signage) for their association with indoor active smoking. The results indicate that no-smoking signs are linked to the absence of indoor smoking. This is a valuable finding, suggesting to authorities that enforcing this specific requirement in HVs can improve compliance with the law's objective. However, the paper misses an opportunity to explore the relationship between other enabling factors mentioned in the law (absence of ashtrays, lighters, designated smoking areas, and shisha equipment) and actual smoking behavior. Understanding the effectiveness of enforcing these additional factors could provide even more targeted guidance for improving compliance.

I recommend expanding the analysis to estimate the association between all five enabling factors and indoor active smoking, further stratifying the results by venue type. This would provide enforcement authorities with more specific, actionable information about which enabling factors are most likely to prevent indoor smoking in different HV types, facilitating the development of more tailored and effective enforcement strategies.

In addition, I offer the following minor comments:

Introduction

• P4 L80: The authors describe the smoke-free requirements of Proclamation No. 1112/2019 as applicable to HVs. L84 specifies that the Proclamation also prohibits using any tobacco product "within 10 meters of any public place or workplace doorway, window, or air intake mechanism." The authors should clarify that the Proclamation prohibits the use of any tobacco product in any outdoor space within that distance.

Methods

• P5 L111: The authors indicate they selected ten major regional and chartered cities in Ethiopia. To understand the study’s results’ applicability, it would be informative to know the percentage of the national population living in these areas.

• P6 L139 & S1: In S1, the authors provide a visual aid to explain the selection of sub-cities, woredas, or kebeles. However, L139 states that they "used these lists (of HVs after mapping) to purposively select 2-4 neighborhoods (or villages),” which seem to be the Primary Sampling Units (PSU). I suggest including the number of PSUs (villages) per previous sampling level (sub-cities, woredas, or kebeles) in S1.

• P6 L142: The authors state that "to ensure representation of various categories of HVs within the chosen kebele/woreda, the assigned sample size for each kebele/woreda was distributed proportionally across the different types of HVs." This suggests that the study did not collect information on all HVs within each PSU or village but rather on the assigned sample size for each kebele/woreda. Please clarify how the sample size was calculated and its value for each kebele/woreda. Also, explain if the assigned sample was distributed proportionally to the number of HVs per type.

• P7 L150: "Data collection tools and procedures": I suggest indicating whether data collectors were instructed to remain for a minimum time at each venue. If not, consider reporting the mean average time of data collection visits in the results section, if you have it. In P22 L396, you discuss that differences between your results and those of other studies “could be due to different observation times and locations”. If "observation times" refers to times of day, also consider including the duration of observation visits as a possible reason for differences, as longer visits increase the chance of observing smoking events.

• P8 L174 Smoke-free compliance indicators: Redraft this section according to the stricter concept of compliance discussed above. Compliance will be measured only through the observance or not of smoking, where prohibited by law. Observing signage, ashtrays, lighters, designated smoking areas, and shisha equipment would measure adherence to other law requirements.

• P8 L180. I am also unclear about why you want to know for indoor compliance if “nobody used tobacco products in the outdoor place within 10 meters from any door, window, or air intake mechanism.” Firstly, tobacco products can include non-combusted products (that don't emit smoke). Secondly, this appears to be the only HV outdoor space where smoking is prohibited by law.

• P8 L183. If I am correct about the prohibition of smoking in the outdoor place within 10 meters from any door, window, or air intake mechanism being the only outdoor no-smoking requirement applicable to HVs, then compliance with the law should only refer to smoking in these areas and the enabling factor requirements only applicable to these limited 10 meters outdoor areas. If the observance of active smoking can be placed within these 10 meter areas, then I would call it compliance. Otherwise, I would not call the indicators about the observation of smoking outdoors and their facilitators “compliance indicators” since the law requires none of the four indicators you propose. I would call them “exposure to SHS outdoors and related indicators.”

• P9 L197. Operational definitions: You define compliance as “the degree to which the HVs fully implement Proclamation No.1112/2019. However, such Proclamation does not seem to ban smoking in all outdoor spaces of HVs. Hence, the clarification I suggest above is whether you are measuring adherence to the law requirements or the law’s direct impact (true compliance) on smoking in indoor and outdoor spaces of HVs.

Also, the law's and the S3 definition of outdoor spaces in hospitality venues is ambiguous, potentially impacting the study's interpretation. The study's limitations section should acknowledge this ambiguity, highlighting how hybrid spaces and temporary structures complicate the categorization of indoor versus outdoor areas. This lack of clarity could affect the accurate compliance assessment with outdoor smoking regulations, potentially influencing the study's findings.

• P9 L206. Data processing and analysis. I find the computation of the overall “average compliance” with the smoke-free laws across all HVs problematic in several ways. First, it includes true compliance and adherence to enabling factors, which should be measured separately. Second, it is not related to specific venues, assuming that adherence is an average. Second, it can mask the underlying non-adherence to key requirements. Suppose your analysis determines that no smoking signage is the most predictive enabler of no smoking in venues. If most venues adhere to every other requirement but not the signage, you may have a high overall average adherence to the law despite a low adherence to the key signage requirement. I suggest dropping overall average adherence by type of HV and city.

• P9 L211. Data processing and analysis. The overall “indoor compliance” metric mixes true compliance and adherence to enabling factors. I suggest splitting the compliance indicator into a true compliance indicator where the primary outcome analyzed is active smoking and adherence to enabling requirements, whose enforcement may best predict the level of smoking indoors and outdoors. This may help the enforcement authority focus on the enforcement efforts to meet the most predictive requirements of success (no smoking).

Results.

Tables. To facilitate reading and comparing table values, arrange the city names and venue types in the same order in all tables.

Redraft the results, including two outcome variables (compliance and adherence to legal enabling requirements), and determine which of the enabling factors best predicts compliance.

Discussion.

P 21 L371.- I suggest restructuring the discussion of the main findings according to the suggested redrafted results.

I suggest expanding the discussion to consider the literature on facilitators and barriers to smokers’ compliance with smoking bans. See for example, doi:10.3390/ijerph13121228 and doi: 10.1136/tobaccocontrol-2017-053920

Conclusions.

P23 L434. I disagree with the author's assessment of good compliance with smoke-free laws in Ethiopia, particularly considering the low prevalence of tobacco smoking in the country. Non-compliance with smoke-free laws should be considered a violation of fundamental human rights and a breach of the WHO Framework Convention on Tobacco Control. In light of this, any instance where more than 5% of venues exhibit smoking in prohibited areas should be considered problematic compliance. Similarly, non-adherence to key enabling factors in more than 5% of venues is equally concerning.

6. PLOS authors have the option to publish the peer review history of their article (what does this mean?). If published, this will include your full peer review and any attached files.

Reviewer #1: No

Reviewer #2: **Yes: **Armando Peruga

---

## [Author Response · Author response to Decision Letter 0]

11 Nov 2024

We have addressed all the reviewer comments and attached as a Respond to Reviewers comments of the point by point responses.

---

## [Decision Letter · Decision Letter 1]

30 Dec 2024

PONE-D-24-30728R1Compliance with smoke-free laws in hospitality venues in Ethiopia: a cross-sectional observational study in 10 citiesPLOS ONE

Dear Dr. Deressa,

Thank you for submitting your manuscript to PLOS ONE. After careful consideration, we feel that it has merit but does not fully meet PLOS ONE’s publication criteria as it currently stands. Therefore, we invite you to submit a revised version of the manuscript that addresses the points raised during the review process. Please submit your revised manuscript by Feb 13 2025 11:59PM. If you will need more time than this to complete your revisions, please reply to this message or contact the journal office at plosone@plos.org. Please include the following items when submitting your revised manuscript:A rebuttal letter that responds to each point raised by the academic editor and reviewer(s). You should upload this letter as a separate file labeled 'Response to Reviewers'.A marked-up copy of your manuscript that highlights changes made to the original version. You should upload this as a separate file labeled 'Revised Manuscript with Track Changes'.An unmarked version of your revised paper without tracked changes. You should upload this as a separate file labeled 'Manuscript'.If applicable, we recommend that you deposit your laboratory protocols in protocols.io to enhance the reproducibility of your results. Protocols.io assigns your protocol its own identifier (DOI) so that it can be cited independently in the future. For instructions see: https://journals.plos.org/plosone/s/submission-guidelines#loc-laboratory-protocols. Additionally, PLOS ONE offers an option for publishing peer-reviewed Lab Protocol articles, which describe protocols hosted on protocols.io. Read more information on sharing protocols at https://plos.org/protocols?utm_medium=editorial-email&utm_source=authorletters&utm_campaign=protocols.

We look forward to receiving your revised manuscript.

Kind regards,

Dereje Oljira Donacho, PhD

Academic Editor

PLOS ONE

Journal Requirements:

Additional Editor Comments:

The study was conducted with two objectives. The first objective was to evaluate the extent of compliance with smoke-free laws in HVs. The second objective was to identify factors associated with non-compliance. The study addressed the objectives with sound methodology and analysis and was well written. However, there are some minor revisions needed to make it more clear for the readers. The authors may have benefited from proofreading the document. I do have a few concerns about the following points: 

1. word usage, like using personal presentation I, we, and ... are not good at using scientific communication. For example, line 127, 'We ----- ... and line 128 ... We considered ... needs revision. I suggest removing the "we" and rephrasing the concept in the context, similar to usage in the document of we,.. which should be corrected.

2. The comparison between the regional towns with AA is another question that should be clarified in the discussion section and justified. 

3. The table titles should be self-explanatory: What, where, and when questions should be answered in the title. e.g., Tables 1, 2, 3, 4, ...?

Reviewers' comments:

Reviewer's Responses to Questions

**Comments to the Author**

1. If the authors have adequately addressed your comments raised in a previous round of review and you feel that this manuscript is now acceptable for publication, you may indicate that here to bypass the “Comments to the Author” section, enter your conflict of interest statement in the “Confidential to Editor” section, and submit your "Accept" recommendation.

Reviewer #2: (No Response)

Reviewer #3: (No Response)

2. Is the manuscript technically sound, and do the data support the conclusions?

Reviewer #2: No

Reviewer #3: Yes

3. Has the statistical analysis been performed appropriately and rigorously? 

Reviewer #2: Yes

Reviewer #3: Yes

4. Have the authors made all data underlying the findings in their manuscript fully available?

Reviewer #2: Yes

Reviewer #3: Yes

5. Is the manuscript presented in an intelligible fashion and written in standard English?

Reviewer #2: Yes

Reviewer #3: Yes

6. Review Comments to the Author

Reviewer #2: See attached file. In my atached comments, I have highlitghed in blue the most important comments that should be addressed by the authors.

Reviewer #3: Topic/title can be rewritten as Compliance with smoke-free laws in hospitality venues and factors associated with non-compliance in.......

This study aims to study/ evaluate the extent of compliance with smoke-free laws in HVs and identify factors associated with non-compliance. But, you don’t have no conclusion about the factors associated with non-compliance. Why?

Can we compare AA with other regional towns? Can we accept the results from these studies? Are they comparable towns/cities? What about the results of the study?

7. PLOS authors have the option to publish the peer review history of their article (what does this mean?). If published, this will include your full peer review and any attached files.

Reviewer #2: **Yes: **Armando Peruga

Reviewer #3: No

---

## [Author Response · Author response to Decision Letter 1]

20 Jan 2025

08 January 2025

06 January 2025

Point-by-Point Responses

PONE-D-24-30728R1: Compliance with smoke-free laws in hospitality venues in Ethiopia: a cross-sectional 

observational study in 10 cities

Reviewers' comments:

Reviewer #2: 

LINE ORIGINAL IN THE DOCUMENT WITH TRACK CHANGES COMMENTS

96-97 However, studies in sub-Saharan Africa have indicated a lack of knowledge and low levels of compliance with smoke-free laws in HVs [8-10]. It is not clear what the “lack of knowledge” refers to. Lack of knowledge of the health effects of SHS? Lack of knowledge of compliance? If this second situation is what it refers to, then it does make sense that studies don´t know what compliance is AND at the same time, they know that compliance is low. Please clarify. 

 Author’s Responses: Thank you for your insightful comments. We have clarified the statement to indicate that the lack of knowledge refers to the requirements of smoke-free laws, leading to low compliance levels among hospitality venue staff in sub-Saharan Africa. We appreciate your thorough review and constructive feedback. 

99 Studies indicate a high prevalence of indoor smoking in HVs [8, 11, 12], Studies where? In Africa? Worldwide? Please clarify

 Author’s Responses: Thank you for your valuable comments and suggestions. To clarify, the studies referenced indicate a high prevalence of indoor smoking in hospitality venues specifically in sub-Saharan Africa, including Uganda, Ghana, and Kenya. We have updated our statement to reflect this geographical context and created a separate paragraph for clarity.

122-123 One of the two studies only studied five government hospitals in Addis Ababa [20] Avoid repetition of the word study for better style. Perhaps write: “One of the two studies was restricted to five government hospitals in Addis Ababa.”

 Author’s Responses: Thank you for your helpful comment. We have revised the sentence to avoid the repetition of the word "study" as you suggested. We appreciate your constructive feedback and the opportunity to improve the style and clarity of our manuscript.

142-143 In this study, we assessed the implementation of smoke-free laws, focusing on two related concepts: true compliance and adherence to the law’s requirements. The two related concepts are compliance and adherence to the law. There is no true compliance as opposed to false compliance. I suggest writing “… focusing on two related concepts: compliance with the smoking ban and adherence to its other requirements.” Please do not use “true” to refer to compliance. 

 Author’s Responses: Thank you for your valuable comments and suggestions. We have revised the sentence to avoid the term "true compliance" and to reflect the concepts more accurately. The updated sentence now reads: "In this study, we assessed the implementation of smoke-free laws, focusing on two related concepts: compliance with the smoking ban and adherence to the law’s requirements. Compliance refers to the law’s effectiveness in preventing active smoking in prohibited areas and serves as a key indicator of its success." We appreciate your constructive feedback and the opportunity to improve the clarity and precision of our manuscript.

195 We then used the comprehensive mapped lists to select HVs, either through systematic or random sampling. Please clarify when you used systematic sampling and when you used random sampling.

 Author’s Responses: Thank you for your insightful comments. We have carefully reviewed your comments and provided additional details to clarify our sampling methods and criteria. Specifically, systematic sampling was used in cities such as Hawassa and Adama, while random sampling was applied in Addis Ababa and Bahir Dar. We appreciate your thorough review and the opportunity to enhance the clarity of our manuscript.

196 In areas where the total number of HVs was small,… You may want to clarify your criteria for a small n. For example, you could write, “In areas where the total number of HVs was small (n<x),…”

 Author’s Responses: In smaller cities like Harar, where the total number of HVs was particularly low, we included all identified HVs in the study. The selection method was based on the local context, and all identified HVs were included when the mapped HVs were less than or equal to the allocated sample size. Therefore, there is no specific 'n' threshold that can be stated. We have not made changes to the original paragraph but have provided this clarification to address your comments.

227 Data collectors observed compliance with the presence of "no smoking" signage at the main entrance and inside the venue,… The use of “compliance” is not appropriate. I suggest rephrasing as “Data collectors observed if "no smoking" signage was properly displayed at the main entrance and inside the venue,...”.

 Author’s Responses: Thank you for your valuable comment. We have accommodated your suggestion and revised the sentence accordingly.

229 & 231 They also noted if anyone was smoking within 10 meters of any main door, window, or air intake mechanism. They also asked the waiter/waitress if ….. For style reasons, avoid repeating “they also” in L231. Perhaps use “In addition they…”

 Author’s Responses: Thank you for your valuable comment. We have accommodated your suggestion and revised the sentence accordingly.

245 9) no one was seen smoking tobacco products in the outdoor area within 10- meters from any door, window, or air intake mechanism (0=no,

1=yes) Remind the reader that indicator nine was obtained even if the venue did not have an outdoor area to serve the public. 

 Author’s Responses: Thank you for your valuable comment. We have clarified that indicator nine was obtained for all HVs, regardless of whether the venue had an outdoor area to serve the public. 

247 The first indicator measures the true compliance with smoke-free law, … As indicated before, I suggest dropping the term true to refer to compliance. I suggest rewriting. “The first indicator measures the compliance with smoke-free law,…”

 Author’s Responses: Thank you for your valuable comment. We have removed the ‘true’ from the statement.

250 Outdoor space compliance: The following five smoke-free specific indicators were used to assess the outdoor space compliance and adherence: Conceptually, these indicators can't measure compliance or adherence to the law because the law does not ban smoking outdoors. I suggest rewriting: “Outdoor Smoking: We also conducted an assessment of smoking-related activity in all outdoor spaces where smoking is permitted. The following five indicators were used: …”

 Author’s Responses: Thank you for your valuable comment. We have clarified that outdoor space refers to any area outside of any HV that is not enclosed but serves the public, where smoking is prohibited. Smoking is not permitted in these outdoor public spaces. The updated sentence now reads: "The following five smoke-free specific indicators were used to assess outdoor space compliance and adherence, defined as any area outside of any HV that is not enclosed but serves the public, where smoking is prohibited: 1).....". A total of 329 HVs included in the sample had such outdoor spaces in addition to indoor spaces. 

332

344 Section on Indoor active smoking and enabling factors

Section on Compliance and adherence with smoke- free laws in indoor areas These two sections contain the same information. The first presents it as noncompliance and nonadherence, and the second as compliance and adherence. I suggest keeping only one, perhaps as “Compliance with the law and adherence to the law requirements in indoor areas,” since Table 2 presents compliance and adherence data. 

 Author’s Responses: Thank you for your valuable comment. We appreciate your feedback regarding the sections on indoor active smoking and enabling factors, and compliance and adherence with smoke-free laws in indoor areas. We would like to clarify the distinction and similarities between these two sections.

In the first section, titled "Indoor active smoking and enabling factors," we aim to provide a detailed textual description of the prevalence of indoor active smoking and the factors that enable it. This section includes data with 95% CIs to highlight the extent of the issue without using tables. This narrative description is essential for understanding the context and underlying reasons for noncompliance. The second section, titled "Compliance and adherence with smoke-free laws in indoor areas," focuses on presenting compliance and adherence data using Table 2. This section provides an assessment of how well the laws are being followed, based on specific indicators.

By maintaining both sections, we ensure a comprehensive understanding of the issue. The first section sets the stage by describing the prevalence and enabling factors of indoor active smoking, while the second section provides a clear and structured presentation of compliance and adherence data. These sections complement each other, offering both a narrative and a data using the Table, which is critical for a thorough analysis. Therefore, we believe that keeping both sections is important for providing a complete picture of the situation. We hope this explanation justifies our decision to retain both sections in the manuscript.

353 Results Make it more straightforward when you talk about compliance and when you do about adherence. 

 Author’s Responses: Thank you for your valuable feedback on our RESULT section. We understand the importance of clearly distinguishing between compliance and adherence in our manuscript. We have made every effort to address your comment by ensuring that each instance of compliance and adherence is presented in a straightforward and distinct manner. Specifically, we have carefully revised the text to avoid any overlap or ambiguity between the two concepts, and have clearly defined each term in the context of our study. Compliance is discussed in terms of how well the HVs follow the smoke-free laws, such as the absence of smoking activity in prohibited areas. Adherence is addressed in terms of the venues' alignment with the broader requirements of the smoke-free laws, such as displaying ‘no smoking’ signage, preventing the presence of smoking-related paraphernalia to maintain SFE.

363 Table 2. Compliance and adherence with smoke-free laws indicators in indoor areas by cities and HVs I suggest tweaking the title of Table 2 and saying, “Compliance with the law and adherence to the law requirements in indoor areas by city and HV type.”

 Author’s Responses: Thank you for your valuable feedback and we have accommodated it.

283 &

377 Data processing and analysis

Table 3. Indoor compliance level with smoke-free laws by cities and HVs Now that you have adopted the concepts of compliance with the law and adherence to its requirements, the classification of HVs into levels of compliance and Table 3 does not make much sense. Compliance should be measured ONLY by seeing active smoking. Then, you could classify levels of adherence to the requirements of the law as ‘fully adherent,’ ‘good adherence,’ ‘moderately adherent,’ and ‘poorly adherent. ‘

I suggest rewriting the piece in the section on data processing and analysis as follows: 

We classified each city and HV type as compliant with the law when no active smoking was detected and as noncompliant when it was detected. In addition, Indoor venues were classified as ‘highly adherent,’ ‘moderately adherent, and ‘poorly adherent’ with the legal smoke-free requirements when they met 7-8, 5-6, and > 5 of these requirements, respectively.

Accordingly, I also suggest rewriting Table 3 with a new appropriate title and only three levels of adherence. Also, rewrite lines 374-376 to describe results and adherence from Table 3.

 Author’s Responses: Thank you for your valuable comments and suggestions. We have revised our manuscript to separately analyze Compliance and Adherence to Smoke-Free law requirements for indoors and outdoors. Specifically, we have made the following changes:

1. Updated the Data Analysis section to clarify the distinction between compliance and adherence.

2. Revised Table 3 and its accompanying text to reflect the new classification of adherence levels.

3. Revised Table 5 and its accompanying text to align with the new approach.

We appreciate your insights, which have greatly improved the clarity and quality of our manuscript. However, we still maintained the classification of Indoor Adherence level by four: Fully compliant (8 points), highly adherent (7 points), moderately adherent (5-6 points) and poorly adherent (<5 points), The frequencies in each category are larger and we wanted to keep the original classification. 

287 & 

417 As indicated before, given that the law doesn’t ban smoking in outdoor areas, it doesn't make sense to talk about compliance with or adherence to the law in these areas. For this reason, II would suppress lines 287-291 in methods and lines 417-418 along with Table 5 in results.

 Author’s Responses: Thank you for your valuable comment. We have clarified above that outdoor space refers to any area outside of any HV that is not enclosed but serves the public, where smoking is prohibited. Smoking is not permitted in these outdoor public spaces. Therefore, we would like to maintain our original descriptions in the manuscript.

383 & 394 Section on Outdoor active smoking and enabling factors

Section on Compliance and adherence with smoke-free laws in outdoor spaces

 These two sections contain almost the same information. The first presents it as negative behavior (smoking) and “no smoking” signage, while the second is as positive behavior (non-smoking) and absence of ashtrays. I suggest keeping Table 4 and retitling it as “Smoke-free indicators in outdoor areas by city and HV type” and combining both sections to describe no-smoking behavior, as well as the other four indicators. 

 Author’s Responses: Thank you for your valuable comment. We appreciate your feedback regarding the sections on outdoor smoking behavior and smoke-free indicators in outdoor areas. We would like to clarify the distinction and similarities between these two sections.

In the first section, titled "Outdoor smoking behavior and enabling factors," we aim to provide a detailed textual description of the prevalence of outdoor smoking and the factors that enable it. This section includes data with 95% CIs to highlight the extent of the issue without using a Table. This narrative description is essential for understanding the context and underlying reasons for noncompliance.

The second section, titled "Smoke-free indicators in outdoor areas by city and HV type," focuses on presenting compliance and adherence data using Table 4. This section provides an assessment of how well the laws are being followed, based on specific indicators such as the absence of ashtrays and the presence of "NO SMOKING" signage.

By maintaining both sections, we ensure a comprehensive understanding of the issue. The first section sets the stage by describing the prevalence and enabling factors of outdoor smoking behavior, while the second section provides a clear and structured presentation of compliance and adherence data. These sections complement each other, offering both a narrative and a data, which is critical for a thorough analysis.

Therefore, we believe that keeping both sections is important for providing a complete picture of the situation. We hope this explanation justifies our decision to retain both sections in the manuscript.

 By the way, the results for the NoDSA indicator are not described in the results section text. Should the NoDSA requirement for outdoor areas be analyzed at all? While a NoDSA requirement for indoor spaces is clearly positive, its value for outdoor areas where smoking is permitted is more controversial. On the one hand, it may transmit that non-smokers need to be protected in DSA outdoors (which is positive for public health). At the same time, because they are ineffective in protecting non-smokers from SHS, they may be considered an excuse not to ban smoking outdoors (which is negative for public health). For these reasons, I wonder if the NoDSA indicator should not be analyzed at all and should be dropped.

 Aut

---

## [Editor Report · Decision Letter 2]

28 Jan 2025

Compliance with smoke-free laws in hospitality venues in Ethiopia: a cross-sectional observational study in 10 cities

PONE-D-24-30728R2

Dear Dr. Deressa,

We’re pleased to inform you that your manuscript has been judged scientifically suitable for publication and will be formally accepted for publication once it meets all outstanding technical requirements.

Kind regards,

Dereje Oljira Donacho, PhD

Academic Editor

PLOS ONE
---

## [Editor Report · Acceptance letter]

30 Jan 2025

PONE-D-24-30728R2 

PLOS ONE

Dear Dr. Deressa, 

I'm pleased to inform you that your manuscript has been deemed suitable for publication in PLOS ONE. Congratulations! Your manuscript is now being handed over to our production team.

Kind regards, 

on behalf of

Dr. Dereje Oljira Donacho 

Academic Editor

PLOS ONE